# Bone Disease in Chronic Kidney Disease and Kidney Transplant

**DOI:** 10.3390/nu15010167

**Published:** 2022-12-29

**Authors:** Ezequiel Bellorin-Font, Eudocia Rojas, Kevin J. Martin

**Affiliations:** 1Division of Nephrology and Hypertension, Saint Louis University, Saint Louis, MO 63103, USA; 2Division of Nephrology and Kidney Transplantation, University Hospital of Caracas, Caracas 1053, Venezuela

**Keywords:** CKD: chronic kidney disease, kidney transplant, bone disease, renal osteodystrophy, CKD-MBD, hyperparathyroidism, low-turnover bone disease, high-turnover bone disease, osteoporosis, PTH, FGF-23

## Abstract

Chronic Kidney Disease–Mineral and Bone Disorder (CKD-MBD) comprises alterations in calcium, phosphorus, parathyroid hormone (PTH), Vitamin D, and fibroblast growth factor-23 (FGF-23) metabolism, abnormalities in bone turnover, mineralization, volume, linear growth or strength, and vascular calcification leading to an increase in bone fractures and vascular disease, which ultimately result in high morbidity and mortality. The bone component of CKD-MBD, referred to as renal osteodystrophy, starts early during the course of CKD as a result of the effects of progressive reduction in kidney function which modify the tight interaction between mineral, hormonal, and other biochemical mediators of cell function that ultimately lead to bone disease. In addition, other factors, such as osteoporosis not apparently dependent on the typical pathophysiologic abnormalities resulting from altered kidney function, may accompany the different varieties of renal osteodystrophy leading to an increment in the risk of bone fracture. After kidney transplantation, these bone alterations and others directly associated or not with changes in kidney function may persist, progress or transform into a different entity due to new pathogenetic mechanisms. With time, these alterations may improve or worsen depending to a large extent on the restoration of kidney function and correction of the metabolic abnormalities developed during the course of CKD. In this paper, we review the bone lesions that occur during both CKD progression and after kidney transplant and analyze the factors involved in their pathogenesis as a means to raise awareness of their complexity and interrelationship.

## 1. Introduction

Chronic kidney disease (CKD) is associated with multiple abnormalities that lead to a complex disorder comprising biochemical alterations in the metabolism of calcium, phosphorus, parathyroid hormone (PTH), Vitamin D, and fibroblast growth factor-23 (FGF-23), abnormalities in bone turnover, mineralization, volume, linear growth or strength, and vascular calcification leading to an increase in bone fractures and vascular disease, which ultimately result in high morbidity and mortality [1,2]. These alterations have been defined by KDIGO (Kidney Disease Improving Global Outcomes) as the Chronic Kidney Disease Mineral and Bone Disorder (CKD-MBD) [2]. Regarding the bone, it has been well established that patients with CKD stages 3a to 5D have increased fracture rates compared with the general population [3,4,5] and that incident hip fractures are associated with substantial morbidity and mortality [6,7,8,9,10]. Kidney transplantation is the treatment of choice for many patients with advanced CKD. However, the improvement of kidney function by means of a kidney transplant, unfortunately does not solve these complex situations completely. Instead, bone disease may persist, progress or transform into a different entity independently of the kidney function due to new pathogenetic mechanisms [11,12,13,14]. In this paper, we will review the bone disease seen in CKD-MBD, also known as renal osteodystrophy, and the pathogenetic mechanisms involved in its development.

## 2. Bone Disease in CKD

The abnormalities of bone in CKD are defined based on bone histology with histomorphometric analysis. Classically, renal osteodystrophy has been classified in different subtypes encompassing a wide spectrum from high turnover to low turnover. At one end of the spectrum, high PTH levels has been considered a surrogate for high-turnover bone disease, known as hyperparathyroid bone disease or osteitis fibrosa, and characterized by elevated bone turnover, increased number and activity of osteoclasts and osteoblasts, variable alterations in osteoid deposition, usually with a woven pattern, and variable amounts of peritrabecular fibrosis (Figure 1B). At the other end of the spectrum, the distinctive pattern is an adynamic bone disease in which typically, bone turnover is decreased with normal mineralization, a paucity of osteoid, bone cells, and a marked decrease in active remodeling sites (Figure 1C). Osteomalacia, another lesion with low turnover, is characterized by increased osteoid seam width, increase in the trabecular surface covered with osteoid, and decrease in mineralization as assessed by tetracycline labeling (Figure 1D). Mixed uremic osteodystrophy is a complex disorder in which elevated bone turnover coexists with features of osteomalacia [15,16,17] (Figure 1E). The frequency of the two last lesions has decreased consistently in recent decades [18]. More recently, it has been shown that osteoporosis is a frequent feature in patients with CKD-MBD [19,20,21] that may complicate their outcome. This disorder, characterized by a decreased bone mass, strength and quality predisposing individuals to bone fracture, is frequent in the CKD population and is likely associated with the elevated risk of fracture not only in the aging groups but also in younger strata [22]. The definition of osteoporosis has been based mainly on bone densitometry. The World Health Organization (WHO) defines osteoporosis as a disease characterized by low bone mineral density and microarchitectural deterioration leading to low bone strength and increase in fracture risk [23]. In the normal population, osteoporosis is defined as a DEXA T score ≤2.5 DS below the normal range for the peak obtained in young persons. In the non-CKD population, both cortical and cancellous bone are substantially reduced [24]. However, in CKD patients, histologic changes are more difficult to interpret as osteoporosis has been seen coexisting with other types of renal osteodystrophy [25] with no difference in prevalence, among them.

As a mean to standardize the definition of bone disease in CKD, KDIGO proposed a new classification based on bone turnover (T), mineralization (M) and volume (V) or TMV, which highlights the most significant bone alterations relevant for clinical evaluation and therapeutic implications [26,27]. However, the classical definition of renal osteodystrophy described above is still used coexisting with the new classification based on TMV.

Most of the early studies describing the CKD associated bone disease were performed in patients with advanced CKD or ESKD (Table 1). In many of them, high-turnover bone disease was described as the predominant form of renal osteodystrophy [17,28,29,30]. More recently, low-turnover bone disease has been increasingly reported in patients with ESKD [31,32]. In some studies, an important number of patients showed normal bone turnover [17,33,34]. In two large series on bone biopsy in patient on dialysis, comprising 630 patients from USA and Europe [34] and 492 patients from different countries (Brazil, Portugal, Turkey, and Venezuela) [35], low-turnover bone disease was observed in 58% and 52% of the patients, respectively (Table 1). Malluche et al. [34] examined the possible role of race in the type of bone lesions occurring in CKD. Biopsies were analyzed using the parameters of TMV as recommended by KDIGO [2]. As a whole, low turnover was observed in 58% of the cases. This type of lesion was more prevalent in white patients, while in black patients, high turnover was observed in 68% of cases. All patients with high bone turnover were younger. Mineralization defects were seen in only 3% of the cases. Regarding bone volume, biopsies in white patients revealed a similar proportion of normal, high or low bone volume, whereas black patients showed a higher proportion of high volume. No differences were observed regarding diabetes, gender, and treatment with Vitamin D. The differences observed in this study suggest that race may be another factor that may influence the type of bone lesion observed in CKD [34].

In the studies summarized in Table 1, patients are heterogeneous with regard to time of the study, gender, age, treatment with vitamin analogs and phosphate binders, calcimimetics, and other drugs prior to the inclusion in the study, as well as the percent of diabetic patients, among others. The use of aluminum containing phosphate binders was relatively common in earlier studies but less frequent in more recent studies. These aspects are important as they may influence bone turnover among other properties of the bone. 

Table 2 summarizes the findings of bone turnover at different stages of CKD not-on dialysis, in studies published from 1976 to 2022. In 1976, Malluche et al. [45] examined bone histology in 50 patients with different stages of kidney disease. It was observed histological evidence of PTH excess, particularly osteoclastic surface resorption, empty osteoclastic lacunae, and woven osteoid in more than 50% of patients with a GFR of 40 mL/1.73 mL/min, whereas endosteal fibrosis was seen when GFR fell below 30 mL/min suggesting that hyperparathyroid bone disease was present since early stages and progressed with advanced CKD. In contrast, in more recent studies, low-turnover bone disease has been increasingly reported predominantly in patients with CKD stages 2 to 4 [46,47,48], with a prevalence that in some cases reached 80 to 100% of the patients. In the study by El-Husseini [48], in patients with a mean eGFR of 44 ± 16 mL/min/1.73 m^2^, low turnover was observed in 84% of the patients. Of interest, most of them had vascular calcifications which correlated positively with levels of phosphorus, FGF-23 and activin, and negatively with bone turnover as has been also reported previously, whereas others have found less prevalence of vascular calcification in lower CKD stages. Other observation of relatively recent studies is the finding of osteoporosis. Nevertheless, the results are not uniform as the prevalence of low-turnover bone varies among studies likely reflecting the different population examined, degree of kidney function, age, ethnicity, geographic distribution, medications including corticosteroids, immunosuppressants, Vitamin D, and type of renal disease leading to CKD, which may play a role in the type and degree of turnover alteration. Indeed, in preliminary studies (*Abstract ASN* 2011) performed in the laboratory of two of the authors in a group of 46 patients with CKD stages 3 to 5, and patients with ESKD on hemodialysis, low bone turnover was observed in 7 (15.2%) of the patients with CKD stages 3 to 5 not on dialysis, whereas high bone turnover was seen in 20 patients (43.5%). Bone alterations consistent with osteoporosis [24,25,34] was found in 12 patients (26.1%). Of interest, this finding was more frequently observed in diabetics (33.5%) compared with non-diabetics (27.3%), female gender, and older age. Normal bone histology was seen in 15.2% of the patients. In contrast, in ESKD, 40 (80%) patients had high bone turnover and only 10 patients (20%) had low bone turnover. In this group, we did not observe changes consistent with predominant osteoporosis or normal bone turnover. There is no clear explanation for the different types of bone turnover predominating at CKD stages as well as the apparent increase in the incidence of low bone turnover lesions along the years reported by different authors. It has been considered that medications, particularly those associated with a possible effect in bone metabolism could explain the differences. Vitamin D analogs and cinacalcet have been shown to affect bone metabolism and particularly bone turnover [49,50,51]. Indeed, in prospective studies, it has been shown that cinacalcet diminishes bone turnover after one year of treatment [51,52]. Thus, it has been argued that oversuppression of PTH by these drugs may result in adynamic bone disease. Our patients with CKD stages 3 to 5 not on dialysis were not receiving treatment with medications that may affect mineral metabolism such as Vitamin Derivatives, calcimimetics, steroids, bisphosphonates, etc. Thus, use of these medications cannot explain the findings. This is similar to that reported in many of studies referred in Table 2. Several other possible factors have been proposed as possible causes of the increase in incidence of low-turnover bone disease. Thus, it has been shown evidence of an indirect relationship between eGFR and turnover, which remains after adjustment for age and the presence of diabetes [48,53]. Likewise, resistance to PTH and an increase in PTH fragments that may have an antagonistic effect [54] is another possibility. Increasing evidence suggest that sclerostin [55,56,57], a factor that increases early during the course of CKD and that is associated with low bone turnover could be one of the main causes responsible for these alterations. Further studies with follow up of patients may be required to obtain an explanation to the apparently changing bone turnover as CKD progresses.

In summary, renal osteodystrophy starts early and progresses along the spectrum of CKD. The histologic pattern has changed along the years towards an increase in low-turnover bone disease in all ranges of CKD. Several factors, including age, gender, race, concurrent disease, use of corticosteroids, diuretics, heparin during hemodialysis, and Vitamin D receptor activators may play a role in the type of bone lesions and vascular disease in patients on hemodialysis. Indeed, it has been shown that Vitamin D agonists influence VDR and osteocalcin expression in circulating in endothelial progenitor cells [61]. More recently, osteoporosis has been considered an important component of bone disease in patients with CKD. 

## 3. Pathogenesis of Bone Disease in CKD-MBD

Figure 2 summarizes the mechanisms involved in the pathophysiology of CKD-MBD. The pathogenesis of CKD-MBD is directly associated with alterations in the regulation of the metabolism of calcium and phosphorus by PTH, 1,25(OH)_2_D_3_, and FGF-23, which closely interact to maintain the homeostasis. 

Extracellular calcium concentrations are tightly controlled within a narrow physiological range optimal for proper cellular functions. Calcium absorption in the intestine is regulated by calcitriol. Calcium acts directly on the parathyroid cell through its specific receptor, the calcium sensing receptor (CaSR), that detects subtle decreases in extracellular calcium, leading to an immediate release of PTH. Conversely, an increase in extracellular calcium suppresses PTH secretion. Numerous studies assign the pathogenesis of the bone alterations of CKD-MBD to the early changes in the metabolism of phosphorus, calcium, FGF-23, and calcitriol that occur as kidney function declines [62,63,64,65,66]. These alterations manifest as an elevation in the levels of FGF-23 and PTH to increase renal phosphate excretion. Studies suggest that FGF-23 increases early in CKD, even prior to a measurable elevation of PTH [67]. It rises phosphate excretion through binding to its klotho co-receptor activating FGFR-1 and FGF-3 receptors leading to a decrease in NaPi2a and NaPi2c cotransporters expression which ultimately will increase phosphate excretion. PTH directly decreases phosphate reabsorption through similar mechanisms. FGF-23 and PTH can maintain phosphate balance until GFR approaches stage 4. Thereafter, neither PTH or FGF-23 are capable of completely maintaining phosphate balance and hyperphosphatemia ensues. The 1,25-(OH)_2_ Vitamin D (calcitriol) synthesis in the kidney is reduced due to the inhibitory effects of elevated FGF-23 and phosphate on 1-alfa- Hydroxylase [32]. Calcitriol deficiency decreases intestinal calcium absorption and diminishes tissue levels of VDR, resulting in resistance to calcitriol-mediated regulation of PTH secretion. The concurrent decrease in the expression of CaSr in the parathyroid cells stimulates PTH secretion [68,69]. In addition, elevated serum phosphorus increases PTH secretion by mechanisms that include a direct action on the CaSr [70]. All these factors in concert lead to the development of secondary hyperparathyroidism. Elevated PTH activate osteoblasts and osteoclasts via the receptor activator of nuclear factor-kappaB (RANK-L) and osteoprotegerin signaling pathway lading to an increase in bone turnover resulting in a bone structure with lower strength and increased fragility, which contributes to an elevation of bone fracture risk and alterations in vascular metabolism resulting in vascular and valvular calcifications [64,65,71,72,73]. Of interest, heparin, an agent to which ESRD patients on hemodialysis are frequently exposed, has been shown to increase osteoprotegerin intra-and postdialytic levels, thus suggesting that this could be an additional factor in the pathogenesis of bone and vascular disease in hemodialysis patients [74]. Additional factors and mechanisms, including inhibition of the canonical Wnt/B catenin signaling pathway, and accumulation of uremic toxins such as indoxyl sulfate have also been proposed as factors that may lead to disruption of the normal regulation of bone turnover leading to renal osteodystrophy and vascular disease [59,75,76,77]. 

The direct role of FGF-23 on bone metabolism is not clear. Studies have shown that in CKD, osteocytes exhibit an increased synthesis of FGF-23. A relationship between FGF-23 and bone abnormalities occurs through its effects on phosphate excretion and suppression of calcitriol synthesis [65,78]. However, FGF-23 levels have been shown to be elevated even before phosphate levels are increased [67]. In addition, it has been suggested an association between FGF-23 with alterations in bone mineralization [79], and FGF-23 and alfa-klotho, the coreceptor for FGF-23, have been shown to stimulate osteoblastic-like cell proliferation and inhibit mineralization [47,78]. Thus, a study in adult dialysis patients found that patients with high bone turnover had higher serum levels of FGF-23 compared with patients with low bone turnover. Likewise, patients with high FGF-23 had normal mineralization, whereas delayed mineralization correlated negatively with FGF-3 levels. Using regression analysis, FGF-23 was the only independent predictor for mineralization lag time [60]. Similarly, in pediatric patients with high bone turnover renal osteodystrophy, it has been shown an association between high serum levels of FGF-23 and improved mineralization, although a correlation between FGF-23 and bone formation rates was not observed [80]. Furthermore, studies examining the expression of proteins in bone tissue of patients with CKD stages 2 to stage 5 on dialysis and healthy individuals have shown that as serum calcium declines, serum alkaline phosphatase, FGF-23, PTH, and osteoprotegerin increase with progression of CKD [47]. These alterations occurred while there was an increase in bone resorption, decreased bone formation and impairment in bone mineralization. Of interest, sclerostin and PTH-receptor-1 expression in the bone was higher in early stages of CKD whereas FGF-23 was elevated in late stages. FGF-23 expression was observed mainly in early osteocytes, whereas sclerostin, which is considered a marker of mature osteocytes, was expressed in cells deeply embedded in the mineralization matrix. These proteins did not co-localize in the same cells. In other studies, high bone turnover was associated with high FGF-23, whereas low bone turnover was observed with lower FGF-23 [81]. Thus, FGF-23 seems to play a direct role in bone metabolism and may be a predictor of bone mineralization in patient with CKD on dialysis and a marker to predict alterations in bone metabolism. 

Studies have demonstrated a relationship between PTH and FGF-23 in the bone. In osteoblast-like UMR106 cells, PTH increases FGF-23 mRNA levels and inhibits sclerostin mRNA messenger, which is an inhibitor of the Wnt/beta-catenin pathway. These studies directly associate the effects of PTH and FGF-23 likely via stimulation of the Wnt pathway. Sclerostin levels vary with renal function. Thus, the expression of sclerostin in jck mouse, a model of progressive kidney disease occurs at early stages of CKD, even before PTH and FGF-23 increase [57], suggesting that sclerostin may play an early role in the development of renal osteodystrophy. Conversely, sclerostin has been associated with adynamic bone disease and vascular calcification [82]. Increased sclerostin has been observed in early CKD but the mechanisms are not clear. It has been suggested that it may relate to partial calcitonin exposure, disturbed phosphate metabolism and extraskeletal sources [55,83]. 

Serum levels of sclerostin and PTH correlate negatively in patients with CKD stage 5D [57]. In addition, in unadjusted and adjusted analysis, sclerostin correlated negatively with bone turnover, osteoblast number and function, and was superior to PTH for the positive prediction of high bone turnover and osteoblast number. On the other hand, PTH was superior to sclerostin for negative prediction of low bone turnover [53]. These findings agree with studies in mice showing that PTH directly inhibits sclerostin transcription in vivo [84,85], suggesting that sclerostin may be useful as a marker of high bone turnover. This negative correlation is in agreement with the negative regulatory function of sclerostin in the intracellular transduction of the PTH signal described in vitro and in vivo [84]. Moreover, studies in patients with renal osteodystrophy subjected to bone biopsy have shown an inverse correlation of sclerostin and PTH levels, as well as a negative association with bone turnover [47,57]. Receiver operator curves analysis showed that PTH and FGF-23 were able to predict high bone turnover, whereas sclerostin was a good predictor of low bone turnover. Sclerostin expression in bone was higher at early stages of CKD and has shown association with lower bone formation rate and greater mineralization defect. These findings favor the notion that sclerostin may play a role in the development of adynamic bone disease in CKD, and hence on fractures [86].

Although elevation of PTH is an early finding in CKD, low bone turnover has been increasingly described in patients with CKD [31,32]. The mechanisms behind these apparently incongruent findings are complex and include maladaptation to the pathophysiologic mechanisms described above, inappropriate PTH signaling and hyporesponsiveness of the PTH receptor, repressed Wnt/B-catenin signaling [55], and elevation of sclerostin in early CKD [47]. Other factors may include the use of medication to prevent or control secondary hyperparathyroidism, oxidative modification of PTH [87], nutritional determinants, age, and underlying diseases. Another factor that has been associated with the early events in the pathogenesis of renal osteodystrophy is indoxyl sulfate. This compound increases in early CKD, even before FGF-23, and has been associated with resistance to PTH [48].

In summary, the mechanisms responsible for the development of renal osteodystrophy are complex and multiple. Based on the sequence of pathophysiologic events that occur during the development of CKD-MBD, high bone turnover is expected to be the most frequent histologic alteration observed in early CKD as PTH starts to increase when GFR drops below 60 mL/min/1.73 m^2^ [10,67,88,89]. Although some studies have confirmed this notion [90], others have shown a high prevalence of adynamic bone disease in the early stages of CKD and even a change of pattern as CKD progresses. Thus, it seems that a single mechanism does not fit all the phenotypes described at the different stages of CKD and that other factors associated with bone disease may also be determinant to the increased bone fragility and fracture in patients with CKD. The interpretation of these findings together is difficult as there is a vast variability in the populations studied, including age, ethnicity, geographic and social background, among others. In evaluating the causes of bone fragility and fracture in CKD, osteoporosis has been increasingly described in patient with CKD and considered to play a major role in bone fracture, particularly given the fact that a high proportion of patients with CKD bone disease have also the usual risk factors for osteoporosis [91]. 

## 4. Bone Fracture in CKD

Bone fracture is a frequent complication of CKD. Table 3 summarizes a number of studies examining the incidence of fractures in patients with CKD. Patients with CKD stages 3a to 5D have higher rate of bone fractures compared with the general population [5,6,7,8,9,10,23,92]. Most studies show that the incidence of fracture increases as GFR decreases, as well as an association between fracture and age in patients with CKD which is superior to that in patients of similar age in the general population. Likewise, the risk of mortality is superior in CKD patients with fracture compared with the general dialysis population. The causes are diverse. Thus, in addition to the pathophysiologic mechanisms leading to renal osteodystrophy, many other factors that include nutrition, medications, concurrent diseases such as diabetes, cardiovascular disease, sarcopenia, increased propensity to fall, among others, may per se increase the risk of fracture in patients with CKD [93]. The estimated prevalence of CKD varies greatly between countries but may reach between 9% and 13.4% of the global population [94,95]. Therefore, the expected prevalence of bone fracture associated with CKD is also high [96]. In most patients, CKD is thought to progress slowly and the bone and the mineral derangement that start in early stages progress relatively silent until it reaches advanced stages in which fracture incidence increases and is more evident. As shown in Table 3, there is an increase in fractures, particularly of the hip [6,7,8,97] but is also seen in other bones. Although this is frequently associated with older age, evidence shows that in general, the incidence of fractures in patients with CKD is elevated. Indeed, as shown in Table 3, several studies have revealed that patients with CKD stages 3 to 5, and those undergoing dialysis have an increased incidence of fractures compared with age-matched subjects without CKD [23]. In addition, the study by Klawansky et al. [98], based on the NHANES III population, a strong trend was noted in which lower bone mineral density (BMD) as determined by dual energy X-ray absorptiometry (DEXA) was strongly associated with reduced creatinine clearance as estimate by the Cockcroft-Gault (CCr) equation. The percentage of women with CCr < 35 mL/min, increased from 0.3% for women with BMD in the normal range to 4% for women with osteopenia to 24% for those with osteoporosis. Similar trend was observed in men, albeit to a lesser extent. Likewise, in another population-based study the prevalence of osteoporosis was 31.8% in patients with CKD stages 3–5 [9]. Thus, the combination of CKD, osteoporosis, and age, results in an elevation of fracture risk.

Osteoporosis is frequent in the CKD population and is likely associated with the elevated risk of fracture not only in the aging groups but also in younger strata [22]. The prevalence of osteoporosis increases with age, a condition that parallels CKD which is also more prevalent in the aging population. Thus, it has been argued that the higher rates of fracture in CKD patients may be a consequence of an increase in the prevalence of age-related osteoporosis. In the younger population, the data are more conflicting. In a recent population based prospective study performed in 19,391 individuals 40 to 69 years old from Canada, with CKD stages 2 and 3 followed for 70 months, it was found that, compared with the median eGFR of 90 mL/min/1.73 m^2^, those with eGFR of ≤60 had an increased risk of fracture in unadjusted and adjusted models [9]. This effect was more evident in younger individual. Hence, there is clear evidence of an increase in fracture risk at all stages of CKD. However, the studies are difficult to compare because most of them followed different methodology and the population studied are heterogeneous. 

The complex alterations that express as low bone mineral density bone mineral density BMD in DEXA studies constitute important components of the bone abnormalities observed in CKD patients and may manifest as osteoporosis and an increased risk for fracture, particularly in those with advanced CKD [104,105,106]. However, in reality, the mechanical properties of the bone express a combination of factors that include aspects related to the specific changes of bone quality associated with renal osteodystrophy [21,107,108] and those due to osteoporosis itself. Thus, low turnover is associated with microstructural abnormalities, whereas bone with high turnover is associated with alteration of material and mechanical property [104]. Some have suggested the combination of the bone alterations described in renal osteodystrophy and the addition of bone fragility osteoporosis be described as “uremic osteoporosis” [65,109]. 

As previously discussed, bone quality in patients with CKD is best characterized by analysis of bone remodeling, mineralizing and volume properties by means of a bone biopsy with histomorphometry. However, this is an invasive method that requires technical and financial resources that are not available for clinical purposes in most centers. In contrast, DEXA, a method widely available, allows the evaluation of bone quantity. Several studies analyzing DEXA in patients with different stages of CKD have provided abundant information indicating an association between CKD-MBD and osteoporosis. In the KDIGO CKD-MBD guideline of 2009 [1], routinely evaluation of BMD testing was not suggested to be performed in patients with CKD, because based on the evidence available at that time, BMD would not predict fracture risk as it does in the general population and would not predict the type of renal osteodystrophy. However, in the updated KDIGO CKD-MBD guideline of 2017 [110], after new prospective studies documented that lower DXA BMD predicts incident of fractures in patients with CKD G3a–G5D [102,111,112,113], it is suggested that in patients with CKD stages G3a–G5D with evidence of CKD-MBD and/or risk factors for osteoporosis, BMD testing to assess fracture risk be performed if results will impact treatment decisions. In one of those studies, BMD by DEXA measured yearly in 485 hemodialysis patients was useful to predict any type of fracture for females with low PTH or to discriminate spine fracture for any patient [111]. Of note, a significant greater risk for fractures was observed with PTH levels either below 150 or above 300 pg-ml. Likewise, bone alkaline phosphatase was a useful surrogate marker for any type of fracture [111,112]. Another study by Naylor et al. [113] has shown that the Fracture Risk Assessment Tool (http://www.shef.ac.uk/FRAX/index.aspx.) with or without BMD measurement is also useful to predict major osteoporotic fracture. Over a period of 5 years, the risk for fracture was not different in individuals with eGFR <60 mL/min/1.73 m^2^ compared with those with an eGFR > 60 mL/min/1.73 m^2^. The trabecular bone score (TBS) is a DEXA-derived algorithm for the evaluation of bone microarchitecture whose utility in patient with osteoporosis has been largely demonstrated [48,114,115]. Studies have shown that TBS may also be useful to measure bone quality in patients with CKD-MBD [114,116]. Indeed, in CKD patient, TBS was significantly associated with the histologic parameters of trabecular bone volume and trabecular spacing but not with dynamic parameters, suggesting that TBS reflects trabecular microarchitecture and cortical width [117]. However, in another study, there was no significant relationship between TBS and bone histomorphometric parameters. TBS has also been shown to correlate inversely with coronary calcification and aortic calcification. Thus, TBS is an important parameter to consider in the evaluation of CKD bone and cardiovascular disease. The fact that TBS may be obtained with DEXA analysis opens an opportunity for further evaluation of bone disease in CKD patients using non-invasive methods. However, further studies are needed to demonstrate whether TBS may predict clinical fractures in patients with CKD-MBD.

Given the limitations of BMD to ascertain the type of bone disease in patients with CKD and the difficulties in performing a bone biopsy as a method to evaluate bone abnormalities in large number of patients, studies have analyzed the possible benefit of other methods than can provide more details of the bone structure. High resolution peripheral quantitative computed tomography (HR-pQCT) can detect microarchitectural changes in patients with CKD [44,118]. A recent study in CKD patients, evaluating bone biomarkers, bone histomorphometry, and HR-pQCT revealed lower BMD, mostly due to trabecular bone impairment compared to controls. It was found that radius BMD and microarchitecture were negatively associated with bone turnover in advanced CKD [44]. HR-pQCT was able to discriminate low bone turnover from non-low bone turnover, whereas there was no difference in DXA BMD between different bone turnover classes. A similar trend has been shown on HR-pQCT and bone turnover as determined by biomarkers in women on dialysis [119]. 

Although several serum biomarkers have been used to assess bone activity in the general population, their use in patients with advanced CKD and those on treatment with dialysis is limited as many of them are affected by renal function. However, recent studies have shown that some biomarkers, are not affected by the kidney function and thus, can be useful to discriminate the type of bone turnover alteration in patients with CKD. As shown in Table 4, PTH, bone specific alkaline phosphatase (BSAP), *N*-terminal propeptide of collagen type 1 (P1NP), and tartrate-resistant acid phosphatase 5B (TRAP-5B), which are not affected by kidney function, may correlate with bone histomorphometry findings. BSAP is produced by osteoblasts during bone formation. Thus, it is considered a marker of bone formation and is associated with fracture and all cause cardiovascular mortality [120,121]. Studies have shown a positive predictive value of BSAP for low bone turnover [35,36,58,122]. In the study by Sprague et al. [35] evaluating the accuracy of bone turnover markers to discriminate bone turnover in 492 ESRD subjected to bone biopsy, PTH (iPTH) and bone specific alkaline phosphatase (BSAP) were able to discriminate low from non-low and high from non-high bone turnover, whereas P1NP, another marker of bone formation, that has been shown to be reliable in CKD in some studies, did not improve the diagnostic accuracy. Similar results, for iPTH and BSAP, have been shown by other studies (Table 4). In contrast, it has also been found that BALP, intact P1NP, TRAP5B, and radius HR-pQCT, but not PTH can discriminate low bone turnover [44]. Other biomarkers such as osteocalcin and FGF-23 albeit promising, require further studies. Therefore, differences still exist as to which biomarkers are useful in the diagnosis of bone turnover. In this regard, Evenepoel et al. [122], in an analysis of several studies on the use of different bone biomarkers, concluded that there is not consistent evidence to replace bone histomorphometry for the diagnosis of bone turnover. It is possible that differences in biomarker assays and the populations evaluated may influence the different results across studies. Meanwhile, the KDIGO CKD-MBD guideline update of 2017 suggests that measurements of serum PTH or BSAP can be used to evaluate bone disease in CKD stages 3–5D because markedly high or low values predict underlying bone turnover stage 3 to 5 [109].

In summary, renal osteodystrophy starts early during the course of CKD and is determined by complex pathophysiologic mechanisms that result in alteration of bone turnover and to a lower extent, mineralization and volume. The pattern of renal osteodystrophy has changed over the years towards an increase in the prevalence of low-turnover bone disease. Osteoporosis is a frequent feature observed in patients with renal osteodystrophy which may play an important role in the elevated risk of fracture in these patients. CKD stage, age, race, and probably other factors may have a role in the type and severity of the bone alterations seen in CKD. PTH and alkaline phosphatase are good predictors to differentiate high and low bone turnover disease, whereas DEXA does not discriminate between the histologic and architectural defects that define renal osteodystrophy but seems important in evaluating the risk of fracture.

## 5. Bone Disease after Kidney Transplantation

For many patients with ESRD, kidney transplantation is the treatment of choice as it improves the quality of life, the metabolic derangements resulting from CKD, and patient survival compared with dialysis. However, numerous studies have demonstrated that after transplantation, bone disease may persist, progress, or evolve into a different phenotype of CKD-MBD [129,130,131,132]. Thus, the risk of fracture in transplant patients increases about 30% over that in CKD patients during the first 3 years after transplantation [99], and up to 25% of transplant receptors may suffer a fracture in the 5 years that follow a kidney transplant. Studies have found evidence of osteoporosis in kidney transplant patients depending on the gender or the DEXA measured site in up to 44% of patients [133]. An interesting aspect to be considered is the relationship between BMD and fracture risk in transplant patients. Although a low BMD is a potent fracture risk factor, many transplant patients with low BMD do not have fractures [134], and an overlap in the BMD measurements between patients with and without fractures has also been shown [135].

Several conditions have been implicated as risk factors for fracture after kidney transplantation. Studies have shown that the risk is increased in white women over 65 years of age, deceased donor kidney recipients, increased HLA mismatches, previous diabetes mellitus, long time on dialysis prior to transplant, the type and severity of pretransplant bone disease, aggressive immunosuppression induction, glucocorticoids, calcineurin inhibitors and alterations in the metabolism of calcium, phosphorus, Vitamin D, and PTH [11,14,132]. Hyperparathyroidism is frequent in CKD as amply discussed previously. PTH levels decrease during the first 3 to 6 months after transplantation [136] due to the improvement of many abnormalities involved in its pathogenesis. Persistent hyperparathyroidism has been considered a maladaptive response of the parathyroid gland resulting from pre-exiting secondary hyperparathyroidism developed during the course of CKD. PTH may remain high particularly within the first year after transplantation in up to 60% of patients [11,14,137,138,139]. The presence of hyperparathyroidism before transplantation, the duration of dialysis, and the development of nodular and monoclonal hyperplasia of the parathyroid gland are the most important mechanisms in the persistence of hyperparathyroidism after transplantation. Therefore, persistent hyperparathyroidism may be a cause of bone disease after transplantation. De novo hyperparathyroidism (compensatory adaptive response) in kidney transplant patients results from elevated PTH levels as a consequence of deteriorating transplant function to maintain phosphate levels and calcium metabolism [140] Therefore, both, persistent hyperparathyroidism and de novo hyperparathyroidism may be a cause of bone disease after transplantation. Hypophosphatemia is a frequent finding after transplantation and has been associated with alterations in bone histology. Mechanisms of hypophosphatemia are diverse, but elevated FGF-23 may play a role. Alterations of Vitamin D metabolism is another important factor that may be associated with posttransplant bone disease as many patients arrive to the transplant period with low levels of Vitamin D [141]. The elevated FGF-23 levels seen in ESRD patients on dialysis decrease after transplantation but may persist high to counteract hyperphosphatemia favored by reduced kidney function. However, at the same time, FGF-23 inhibits inhibit 1-alfa-hydroxylase in the kidney resulting in lower levels of 1,25(OH)_2_Vitamin D. These alterations, together with elevated PTH favor the development or maintenance of the bone lesions observed in kidney transplant recipients.

Age has been shown to be a strong risk factor for fracture. Thus, for each decade of life, the risk of hip fracture is estimated to be 55% higher. Fortunately, the role of glucocorticoid use, an important risk factor for bone fractures, seems to be decreasing in transplant patients after new protocols with lower doses or shorter term are increasingly common [142,143,144,145].

Initial studies showed that the changes in BMD differ in early and late posttransplant periods. Julian et al. [136] showed that BMD in the lumbar spine decreases sharply by almost 7% during the first year after transplantation with a somewhat less pronounced decrease thereafter, reaching around 9% at 18 months. Milkus et al. [146] in a prospective study showed a loss of BMD in lumbar spine within 6 months with no significant loss in femoral neck, while other authors have reported a decrease in BMD in femoral neck at 3 months after transplantation [147]. The bone loss seen after a kidney transplant may persist for years but ultimately tends to improve in patients with preserved kidney function. Carlini et al. [11,12,148] demonstrated in a group of long-term renal transplant patients that BMD progressively improved as time after transplantation increased, approaching normal values after 10 years. Other studies analyzing BMD in patients between 6 and 20 years after renal transplantation showed a mean annual decrease in lumbar T scores of −0.6 ± 1.9% [134], a value relatively similar to that observed in the general population with aging [149]. In an evaluation of 3992 first kidney transplants from the Swedish National Renal Registry, 279 fractures occurred (7% of all patients), of which 139 were in the forearm, 69 in the hip, 45 in the humerus and 26 in the spine. The multivariate-adjusted fracture incidence rates were highest during the first 6 months after transplantation, and 86% higher in women than in men. High age, female gender, diabetes nephropathy, history of previous fracture, dialysis, and acute transplant rejection were associated with an increased risk of fracture [150], whereas pre-emptive kidney transplant was associated with a lower risk of fracture. In patients with major fractures followed by a median time of 3 years, mortality associated with hip fracture was the highest (148/1000 patient-years) followed by spine fracture (87/1000 patient year). In adjusted Cox proportional multivariate analysis hip fracture and spine fractures had the worst clinical outcome, whereas forearm fractures were not associated with increased mortality risk [150]. 

Recently, Evenepoel et al. [145], examined prospectively BMD and bone turnover markers in transplant recipients. BMD was determined by DEXA within 2 weeks after transplantation. During the average follow-up of 5.2 years, 38 patients (7.3%) sustained a fragility fracture, corresponding to 14.2 fractures per 1000 person-year. The median time from transplantation to the first fracture was 17.1 months, and the prevalence of osteoporosis ranged between 10% and 35%. In more detail, osteopenia and osteoporosis were 34.5% and 39.6% at the distal radius, and 22.1% and 55% at the femoral neck, respectively. Patient with lower T-scores were women, older, had higher PTH and lower BMI and sclerostin. Of interest, bone turnover markers were inversely correlated with BMD at all skeletal sites, suggesting that bone markers may increase the ability to evaluate fracture risk in this patient population. Of note, 30 patients from a total of 518 evaluated, had a previous history of fragility fractures at the time of transplantation, and osteoporosis or osteopenia was observed in 77% of them, indicating that they had a severe bone disease prior to transplantation. In a more recent prospective study by the same group performed in 97 subjects evaluated before or at the time of kidney transplantation, and re-evaluated at 12 months posttransplant, it was observed that changes in BMD were highly variable ranging from −18% to + 17% per year. Significant bone loss was observed at the distal third radius and ultradistal radius, with no overall changes in BMD at the spine or hip. Cumulative steroid dose was related with bone loss at the hip, while resolution of hyperparathyroidism related to bone gain at spine and hip [151]. It is important to consider that the incidence and type of fracture varies between different studies, which may reflect the heterogeneity of the populations analyzed. It seems that fractures at peripheral sites rather than central skeleton are more frequent [150]. 

In conclusion, decreased BMD is frequent and more pronounced during the first months after kidney transplantation but varies individually and with the population studied. Age, sex, time in dialysis before transplantation, bone disease before transplantation, steroids use, and mineral metabolism, among others are important risk factors for the loss of BMD observed, and ultimately for fracture and mortality. 

## 6. Alterations of Bone Histology after Transplantation

In early reports, posttransplant bone disease was mainly considered a consequence of glucocorticoids. However, it has become evident that it comprises a variety of metabolic alterations in the pretransplant and posttransplant periods. The pre-existing renal osteodystrophy plays a central role as this is an almost universal complication in patients with CKD. Thus, as previously described, different alterations in bone turnover, volume, and to a lesser extent, mineralization are the dominant features of altered bone histology in CKD patients. In addition, many patients with CKD arrive to transplant with different degrees of osteoporosis and alterations of bone strength. Although after transplantation, the rapid improvement of kidney function helps controlling many of the pathophysiologic mechanisms involved in the development of renal osteodystrophy, in many cases kidney function does not reach normality or decreases after some time, thus maintaining to a certain degree part of the known mechanisms responsible for bone alterations in CKD. In other cases, bone alterations such as osteoporosis may occur or progress adding to the basic defective bone metabolism.

The histology of the bone alterations described after transplantation help understand its pathogenesis and evolution along the years. The initial studies by Julian et al. [136], showed that as early as 6 months after transplantation, patients showed a decrease in bone formation and bone apposition rate, and reduced cortical thickness. These findings were interpreted as adynamic bone disease. The factors involved in these changes may include a rapid decrease in PTH levels in patients with relatively mild bone disease, and the effect of high doses of glucocorticoids used in early months after transplantation. Similarly, in patients undergoing bone biopsy around 6 years after transplant, low bone turnover, low bone volume and focal or generalized osteomalacia were frequent histologic features [152]. In patients with long-term renal transplantation and relatively preserved GFR the results are discrepant. We have described a mixed histologic pattern with increased bone resorption, low bone formation, and prolonged mineralization lag time in patients with a mean of 7.5 years after transplantation and relatively preserved kidney function. These lesions were more severe in patients with less time after transplantation but approached normal values in patients with more than 10 years after transplant [148,153]. Similar results have been published by others [13]. Other studies have confirmed that the pattern of histologic changes after transplantation is heterogeneous although a decrease in bone formation and mineralization tend to be a relatively frequent finding [154,155,156,157]. Prospective studies performed at different times posttransplant have helped to clarify the changes in bone histomorphometry. Rojas et al. [158] studied the early alterations in bone remodeling in patients subjected to a bone biopsy performed on the day of transplantation and repeated within 21 to 120 days later. The main histologic alterations were a decrease in osteoid and osteoblast surfaces, and a decrease in bone formation rate with a prolongation of mineralization lag time. In addition, almost half of the patients, most of them with adynamic bone disease and mixed bone disease prior to transplant, showed early osteoblast apoptosis and a decrease in osteoblast surface and number. Of interest, lower levels of serum phosphate were observed in those patients whose biopsies showed osteoblast apoptosis, and there was a positive correlation between posttransplant serum phosphate and osteoblasts number. It was suggested that impaired osteoblastogenesis and early osteoblast apoptosis may play a role in posttransplant bone disease. Hypophosphatemia has been described in the posttransplant period, but it is not clear to what extent it may be associated with mineralization defects [155]. In a recent prospective longitudinal study, Jorgensen et al. [151] showed that phosphate levels and nadir after transplantation correlated inversely with changes in mineralization, suggesting that hypophosphatemia may negatively affect bone mineralization. PTH levels, glucocorticoids, and phosphatonins have been suggested as possible causes of hypophosphatemia in these patients [137]. 

Analysis of bone biopsies utilizing the TMV system have helped understand and define the histologic changes that occur in the bone after transplantation. In a recent prospective study, Keronen et al. [157] showed that the proportion of patients with high bone turnover declined from 63% to 19% at two years posttransplant, while the presence of low bone turnover increased from 8% to 38% and mineralization defects increased from 33 to 44%. Trabecular bone volume showed little changes after transplantation. In a prospective study involving 97 patients undergoing a bone biopsy before or at the time of transplantation and a repeat biopsy 12 months after transplantation [151], bone turnover remained normal or improved in the majority of patients. Bone resorption as well as fibrosis decreased while delayed mineralization was present in 15% of the patients after transplantation. Hypophosphatemia was an important finding, particularly during the first 3 months. However, at 12 months 13% of the patients still had hypophosphatemia (serum phosphate less than 2.3 mg/dL). Interestingly, delayed mineralization was observed in a subset of patients after transplantation and associated with the duration and severity of hypophosphatemia. In densitometry analysis, the cumulative dose of steroids was related to bone loss at the hip whereas resolution of hyperparathyroidism was related to bone gain.

Thus, bone disease after transplantation is a complex disorder that relates to the pretransplant renal osteodystrophy, changes in mineral metabolism after transplantation, ethnicity, age at the time of transplantation, prior and de novo acquired hypoparathyroidism, osteoporosis, use of glucocorticoids, and CNI inhibitors among other factors. Hence, in analyzing bone disease after kidney transplantation, it is important to consider that not all patients arrive to transplant with a uniform bone histologic pattern. High bone turnover has been observed in many of them, whereas in other studies, low bone turnover has been the most frequent finding. Interestingly, however, since the early studies a decrease in bone turnover after transplantation seems to be the most frequent observation. Similarly, decreased mineralization has also been reported in several studies. The mechanisms responsible for this change are not completely clear. It is possible that impaired phosphate metabolism plays a role in this bone defect, at least in part, given the association of impaired mineralization with low phosphate levels observed in a subset of transplanted patients.

An additional consideration should be given to discrepancies that may occur in the interpretation of the findings of the bone biopsy as the reference ranges used by different laboratories for histomorphometric parameters may vary. Indeed, in a recent reanalysis of two set of data of bone histomorphometry in kidney transplant patients using reference ranges from different laboratories, there were disagreements in the categorization of bone turnover on the basis of the cutoff applied, which at the end may affect the diagnostic precision with potential clinical impact [159].

In summary, CKD-MBD [2] comprises mineral and hormonal alterations, abnormalities in bone turnover, mineralization, volume, linear growth, leading, and vascular calcification leading to an increase in bone fractures and vascular disease, which ultimately result in high morbidity and mortality. The bone alterations occurring during the course of CKD, typically described as renal osteodystrophy may extend or evolve to a different pattern after kidney transplantation by mechanisms that include but are not limited to those leading to the CKD. Osteoporosis is a frequent finding in patients with CKD-MBD that can aggravate the bone disease and increase the risk of fracture (Figure 3).

## Figures and Tables

**Figure 1 nutrients-15-00167-f001:**
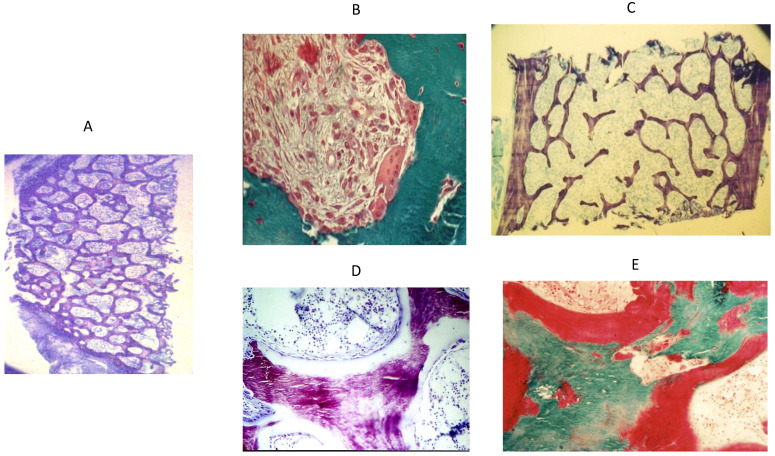
Examples of different types of bone disease in CKD Bone histology in CKD. (**A**) Panoramic view of normal bone histology. Magnification 4×. Toluidine blue staining. Figure shows interconnected bone trabeculae (purple). Clear areas between trabeculae represent medullary space. (**B**) Hyperparathyroid bone disease (high bone turnover). Magnification 20×. Goldner trichrome staining. Figure shows a resorption area with multinucleated osteoclast in the periphery. Area in blue corresponds to a calcified trabecula. (**C**) Adynamic bone disease (low bone turnover). Magnification 20×. Toluidine blue staining. Figure shows a bone section without cellular activity. Trabeculae are thin and disconnected. (**D**) Osteomalacia (low bone turnover). Magnification 20×. Toluidine blue staining. Figure shows abundant osteoid matrix (light blue) covering the mineralized bone trabecula. (**E**). Mixed uremic osteodystrophy (combines features of high turnover and osteoid matrix). Magnification 20×. Goldner trichrome staining. Picture shows an area of osteoclast resorption with abundant osteoid covering the bone trabecula.

**Figure 2 nutrients-15-00167-f002:**
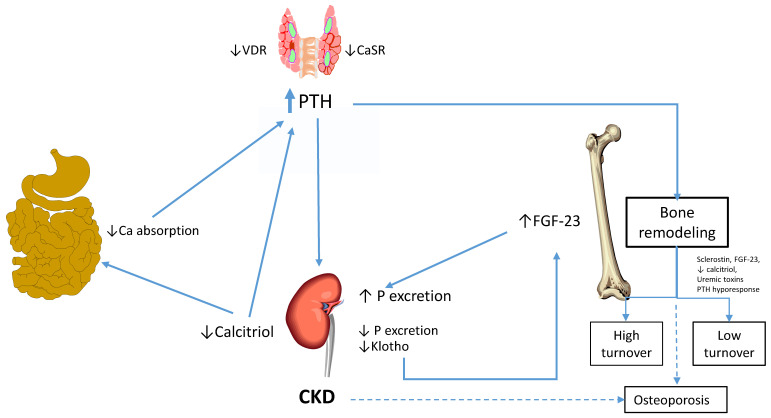
Summary of the pathophysiology of CKD-MBD. The progressive loss of kidney function results in a decrease in renal phosphate excretion. FGF-23 produced by osteocytes and osteoblasts increases early in CKD decreasing NaPi2 and 2c cotransporters expression in the kidney leading to phosphaturia. PTH decreases phosphate reabsorption by similar mechanisms. At some point during the progression of CKD, when GFR approaches stage 4, both mechanisms are insufficient to maintain phosphaturia efficiently and serum phosphorus rises. In addition, FGF-23 inhibits 1-alfa-Hydroxylase synthesis of 1,25-(OH)_2_ Vitamin D (calcitriol) in the kidney, which results in a decrease in intestinal calcium absorption, serum calcium concentration, and reduction in tissue VDR, resulting in resistance to calcitriol-mediated regulation of PTH secretion. With time, hyperplasia of the parathyroid glands ensues. If conditions persist, hyperplastic glands transform into nodular hyperplasia and lately, into single nodular glands. These glands have a decreased expression of CaSr and consequently, poor response to calcium and calcimimetics. All these factors in concert, lead to the development and progression of secondary hyperparathyroidism. PTH activates osteoblasts and osteoclasts leading to an increase in bone remodeling. Typically, bone turnover in patients with CKD may be elevated as a consequence of the action of PTH in the bone or decreased due to mechanisms not completely understood that include among others, elevated sclerostin, FGF-23, uremic toxins, low calcitriol levels, and hyporesponsiveness of the cells to PTH. In addition to the typical lesions classically described in CKD, osteoporosis has been increasingly described in CKD and seems to play an important role in the elevated risk of fracture in these patients.

**Figure 3 nutrients-15-00167-f003:**
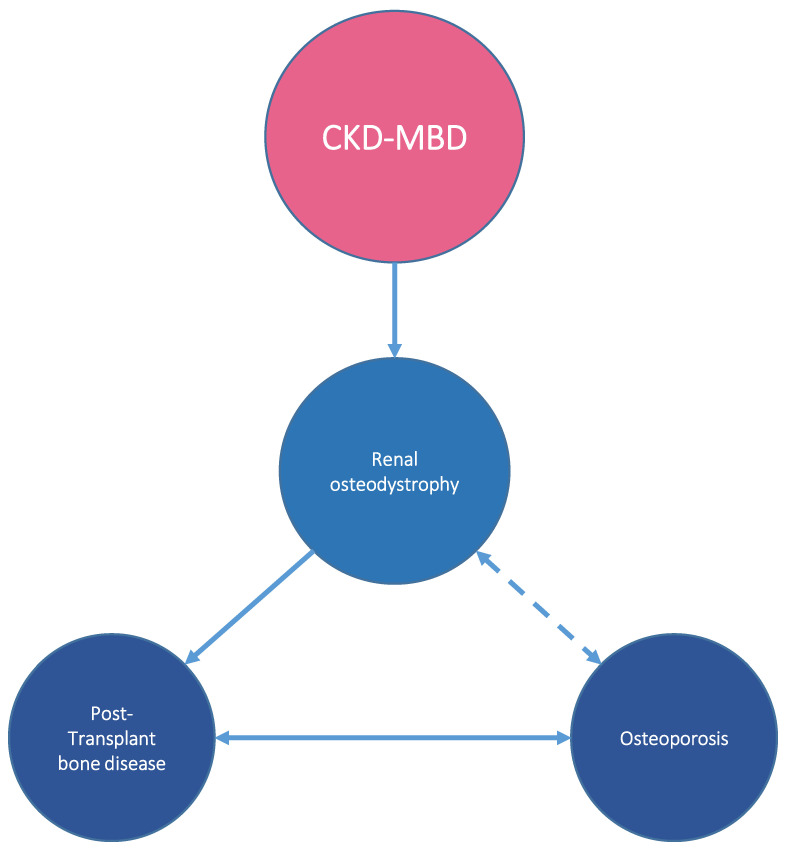
The bone alterations in CKD-MBD, typically described as renal osteodystrophy, may extend or evolve to a different pattern after kidney transplantation by mechanisms that include but are not limited to those leading to the CKD. Osteoporosis is a frequent finding in patients with CKD-MBD that can aggravate the bone disease and increase the risk of fracture.

**Table 1 nutrients-15-00167-t001:** Type of CKD-associated alteration of bone turnover (BTO) in patients with advanced CKD (ESKD or pre-dialysis).

Reference	Country	Patients Number	Age Years	Kidney Function	Low BTO % Patients	High BTO % Patients	Normal BTO % Patients	Comments
Couttenye [36] 1996	Belgium, Greece, Egypt, Argentina, Slovakia, Luxemburg	103 Female 53	59.7 ± 1.3	ESKD	46.7	40.8	12.6	P P-binder: 89 pts (CaCO_3_, 33, Al (OH)_3_ 31, both 31). Vit D: 42 pts. Al overload: 12 pts PTX: 8 pts. Previous kidney transplant: 8 pts
Salusky [17] 1988	USA	44 Fem 22	11.8 ± 5.8	ESKD CAPD	20	64	16	P-binder: 45% of pts on Al (OH)3. CaCO_3_ 54.5% of pts. All pts receiving PO calcitriol Bone surface AL in 10 out 20 pts receiving Al (OH)_3_.
Rodriguez-Perez [37] 1992	Spain	26	NA	ESKD	57	43	NA	NA
Sherard [38] 1993	USA 239 Canada 20 pts	259 Female 63	PD: 60 ± 12 Female 63, HD: 52 ± 1.4 Female 39	ESKD	66 38	44 62	NR	P-binder: Al (OH)_3_ predominantly early in the study. CaCO_3_ predominantly later. Vitamin D: 12.7% of the pts). Surface bone Al 25% in 31% of the pts with low BTO. Serum Al higher in HD. Low BTO more frequent in PD. Diabetes: 30% of pts on PD and 19% of pts on HD PTX: 4% of pts on PD, 6% of pts on HD. Dialysis fluid Ca concentration 3.25 meq/L
Torres [39] 1995	Spain (Canary Islands)	119 Female 80	47 ± 16	Predialysis 38 HD: 49 CAPD: 32	60 44 59	40 56 61	NR	P-binder: CaCaO_3_ was primary. Al (OH)_3_ added to 40% of predialysis pts and 56% of dialysis pts.). Vitamin D (PO alfa-calcifediol): HD: 6, CAPD 12, predialysis none
Hamdy [40] 1995	Belgium, France, Netherlands, United Kingdom.	176 *	18–81	CrCl 15–50 mL/min	4 7	91 93	NA	* Placebo 87 Alfacalcidol 89
Monier-Faugere [15] 1996	USA	2248 Female 1144	Female 54 ± 0.4 Male: 50 ± 0.5	ESKD HD: 81.4% CAPD 18.6%	22.6	77.4	NA	P-binder: Al containing more used between 1983 and 1987. Ca-based binders use increased progressively. Vitamin D: Calcitriol: 20% of pts; use of IV progressively increased in time. Al staining at 30% of trabecular surface in 85% of pts. Trend decreased progressively. BTO type varied within the study period. Proportion of use of DFO decreased along the survey.
Coen [41] 2002	Italy	Predialysis 79 HD 107	51 ± 14 51.1 ± 12	S. creatinine 5.2 ± 2.5 ESKD	Predialysis 21.5 HD 14	Predialysis 65.8 HD 86	Predialysis 12.6 NA	P-binder or Vitamin D: Predialysis patients: No. HD pts: CaCO_3_ Ca None of the pts were on Vitamin D. Bone Al higher in HD pts.
Coen [42]	Italy	41	51.9 ± 12.4	ESKD HD	22	78		P-binder: Ca-based in most patients. Al agents in 39% of pts for short periods. Calcitriol in a minority of pts.
Spazovsky [31] 2003	Macedonia	84 Female 40	54.2 ± 12	CCl < 5 Non-dialysis	35	27	38	P-binder: CaCO_3_ in 70% of pts). Vitamin D analog: None of the pts.
Ferreira 2008 [43]	Portugal	68 Female	55. ± 15.4 * 53.9 ± 13.7 **	ESKD	* 67 > 56 ** 60 > 54	33 > 46 40 > 46	NR	P-binders: * Sevelamer N = 33. ** Ca-based N = 35. Pts underwent bone biopsy prior to and at the end of treatment with either P-binder. BTO is shown at both periods.
Malluche [34] 2011	USA Europe	630 Black 87 White 543	55 ± 1 Black 50.7 ± 1.4 White 56.3 ± 0.6	ESKD HD: 600 PD: 30	58	24	18	Black patients were younger, had less time on dialysis, more treatment with vit D analog and non-Ca/non-Al containing P-binders. Diabetes prevalence: 25.3% black vs. 20.4% white pts, respectively. P-binder: CaCO_3_ 429 pts. Active Vitamin D analogs: 109 pts. Dialysate Ca: 2.5 meq/l in 371 pts, Ca 3.3 meq/l in 259 pts.
Sprague [35] 2016	Brazil Portugal, Turkey Venezuela	492 Female 218	49.5±	ESKD	52	48	MR	P-binders: Ca-based 379; Al salts 52, Sevelamer 42 pts, respectively. Vit D 143 pts. Corticosteroids 24, immunosuppressives 20 pts. Previous kidney transplant 46. Parathyroidectomy 5.
Salam [44] 2018	UK	43	59 ± 12	CKD 4–5 65% predialysis	26	40	34	Diabetes: 28% of pts vs. 0 in controls. Previous fragility fracture 22% of pts vs. 7% in controls. By HR-pQCT, CKD pts had lower BMD, trabecular thickness, and trabecular bone volume at distal radius and distal tibia compared with controls.
Carbonara [25] 2020	Brazil	260	51 ± 12		21.6	76.6	1.8	Osteoporosis in 35% of pts irrespective of ROD type. Al staining in 38% of the biopsies.

BTO: bone turnover; ESKD: end stage kidney disease; P-binder: phosphate binder; PO: oral; HD: hemodialysis; PD: peritoneal dialysis; CAPD: continuous ambulatory peritoneal dialysis; CCl: creatinine clearance; pts: patients, VC: vascular calcification; * refer to placebo. ** refers to alfacalcdol.

**Table 2 nutrients-15-00167-t002:** Distribution of types of CKD-associated alteration of bone turnover (BTO) in patients at different CKD stages (not on dialysis).

Reference	Country	Patients Number	Age Years	Kidney Function CKD Stage or GFR ml/min/1.73 m^2^	Low BTO % Patients	High BTO % Patients	Normal BTO % Patients	Comments
Malluche 1976 [45]	Germany	50 Female 31	43.2. ± 11	6–80	NA	Most patients	Not reported	BTO increased with decreasing GFR. None of the pts were receiving vit D analogs, PO phosphate binders or Ca supplements.
Bervoets [58] 2003		84		Predialysis ESKD	35	27	38	Active Vitamin D in 30.8% of the pts across CKD stages, mostly CKD stages 4 to 5D. Calcium containing P-binder in 21.2% of pts, mostly in CKD stage 4 to 5D
Tomiyama [46] 201	Brazil	50 Female 66		GFR 15–90	CKD 2: 100 CKD 3: 88 CKD 4: 78	0 0 2	0 12 20	No treatment with P-binders or vit D analogs. High prevalence of hypertension, diabetes, overweight/obesity dyslipidemia. CAC detected in 66% of pts correlated with bone turnover.
Barreto [59] 2014; Drueke [32] * 2016	Brazil	49 Female 32	52	CKD stage 2–5 GFR 36 ± 17	CKD 2–3: Predominant low BTO	CKD 4–5: Predominant high BTO	Not reported	No P-binder or vit D treatment Association of indoxyl sulfate with osteoblast surface and bone fibrosis * Reanalysis of the same data
Lima [60] 2019	USA	104 Female 75	59 ± 15	CKD stage 2: 22 pts CKD stage. 3: 29 pts CKD stage 4–5: 19 pts ESKD HD: 34 pts	55	33	13	Treatment with active Vitamin D in 30.8% of the patients across the different stages of CKD, mostly in CKD st 4 to 5D Calcium containing P-binder in 21.2% of patients, mostly in CKD stage 4 to 5D
Graciolli [47] 2017	Brazil	148 Female 51	50–54	CKD stage 2–5 CKD stage 2–3 CKD stage 4 CKD stage 5	83 94 83 81	17 6 17 19	Not reported	P-binder or Vitamin D: Predialysis patients: No. HD pts: CaCO_3_ Ca None of the pts were on Vitamin D. Bone Al higher in HD pts. Predictive value of iPTH is higher in HD in HD.
El-Husseini [48] 2022	USA	32	61 ± 11	44 ± 16	84	16	Not reported	Calcium supplement 2 pts, Vitamin D none. Diuretics 15 pts. In white pts, eGFR correlated negatively with BTO. Most pts had VC >80%. VC correlated positively with serum P, FGF-23, and activin. TBS correlated negatively with coronary calcification

BTO: bone turnover; ESKD: end stage kidney disease; BALP: bone alkaline phosphatase; P-binder: phosphate binder; CAC: coronary artery calcification; ALP: alkaline phosphatase; TBS trabecular bone score.

**Table 3 nutrients-15-00167-t003:** Incidence of fractures in CKD.

Reference	Cohort Number of pts (N)	CKD Stage/CCl ml/min or eGFR ml/min/1.73 m^2^	Fracture Type and Number	Fracture Incidence 1000 Person-Years	Comments
Alem [5] 2000	USRDS data base N: 326,464 person-year	ESKD	Hip 6542	Men 7.45 Women 13.63	Relative risk highest in younger people. Added incidence of fracture increased with age and was greater for women than for men.
Coco [6] 2000	N: 4039 person-year, N: 1272 pts treated	ESKD HD	Hip 56	Men 11.7 Women 24.1	The one-year mortality rate from hip fracture was ~2.5 times higher in dialysis pts compared with general population.
Jadoul [99] 2006	DOPPS: HD pts. 12,782	ESKD HD	Hip 174 Any 498	8.9 for hip 25.6 for any new fracture	Older age, female sex, prior kidney transplant and low serum albumin were predictive of new fracture. PTH > 900 was associated with risk of new fracture
Danese [100] 2006	DMMS data base N: 9007 pts	NA	Hip and vertebral	580/1000 vs. 217/1000 in the general dialysis population	Age and sex-adjusted mortality rate after fracture 2.7 times greater than the dialysis population. Pts with lower PTH were more likely to sustain a hip fracture than those with higher PTH.
Dukas [101] 2005	Cross sectional N: Women 5313 N: Men 3238	CrCl ml/min 60.9% < 65 39.1% ≥ 65	Not reported	Not reported	CCl < 65 increased risk of experiencing falls and risk for hip fracture (OR 1.57, 95%CI 1.18–2.09, *p* = 0.002), and for vertebral fracture
Lin [8] 2014	Taiwan NHIRD N: 51,473 incident dialysis patients	ESKD Dialysis	Hip 1903	8.92 Men 7.54 Female 10.12	HD pts had a 31% higher incidence of hip fracture than PD patients (HR 1.31, 95% CI: 1.01–1.70). Patients ≥65 years old had more than 13 times the risk of a hip fracture than those 18–44 years old (HR: 13.65; 95% CI: 10.12–18.40)
Tentori [90] 2014	International DOPPS N: 34,579	ESKD/HD	NA	Japan 1 Belgium 45	Fracture pts had 3.7-fold higher rates of death compared to DOPPS population. In most countries, mortality rates exceeded 500 per 1000 patient-year
Naylor [97] 2014	Data base from Ontario, Canada N: 679,114	eGFR ml/min/1.73 m^2^: ≥60; 45–59; 30–44; 15–29; <15	Hip Forearm Pelvis Humerus	Not available in women ≥ 60	Fracture rate in women ≥ 65 years old at different eGFR (ml/min/1.73 m^2^): >60: 4.3% 45–59: 43%: 5.8% 30–44: 47.9%: 6.5% 15–29: 54.4%: 7.8% <15.54.2%: 9.6
Naylor [102] 2015	2107 320 individuals with eGFR < 60 mL/min/1.73m^2^ 1787 individuals with eGFR ≥ 60 mL/min/m^2^	eGFR ml/min/1.73 m^2^: ≥60; 45–59; 30–44 15–29; <15	64 (3%) over 4.8 years	Not available	The 5-year observed major osteoporotic fracture risk was 5.3% in individuals with eGFR < 60 mL/min/1.73m^2^ was 5.3%, comparable to the FRAX predicted fracture risk. No difference in the AUC values for FRAX in individuals with eGFR < 60 mL/min/1.73 m^2^ vs. those with eGFR ≥ 60 mL/min/m^2^
Yamamoto [103] 2015	3276			* 1.48 ** 2.33	Mortality was lower in pts * using ACEI/ARB than those ** not using ACEI/ARB 13.6% vs. 16.8%
Hung [7] 2017	Taiwan’s NHIRD Total of 61,346 first fragility hip fracture nationwide. 997 dialysis hip fracture patients were matched to 4985 non-dialysis hip fracture subjects	ESKD Dialysis	Hip 997	Not available	Higher proportion of femoral neck fractures in the dialysis group compared to the non-dialysis group (51% and 42%, respectively; *p* < 0.001) The mortality rate was significantly higher for patients in the dialysis group, with a mortality rate of 91% compared to 71% for those in the non-dialysis group ( *p * < 0.001).
Desbiens [9] 2020	CARTaGENE data base (CAG) N: 679,114 19,391pts with CKD included	Non-CKD: 9521 CKD: 2: 9114 CKD 3: 756	829 Various type	Non-CKD: 6.9 CKD 2: 7.6 CKD 3: 11.3	Compared with the median eGFR of 90 mL/min/1.73 m^2^, eGFRs of 60 mL/min/1.73 m^2^ were associated with increased fracture incidence [adjusted hazard ratio (HR) ¼ 1.25 (95% confidence interval 1.05–1.49) for 60 mL/min/1.73 m^2^; 1.65 (1.14–2.37) for 45 mL/min/1.73 m^2^]. The effect of decreased eGFR on fracture incidence was higher in younger individuals [HR 2.45 (1.28–4.67) at 45 years; 1.11 (0.73–1.67) at 65 years and in men.

URDS: United States Renal Data System; DOPPS: Dialysis Outcomes and Practice Patterns Study; Taiwan NHIRD: Taiguan’s National Health Insurance; HD: hemodialysis; CCl: creatinine clearance. FRAX: Fracture Risk Assessment Tool.

**Table 4 nutrients-15-00167-t004:** Relationship between bone markers and discrimination of the type of bone turnover alteration in CKD.

Reference	Country	Number of Patients	Age Years	Kidney Function	Comments
Coutteneye [36] 1996	Belgium, Greece, Egypt, Argentina, Slovakia, Luxemburg	103 Female 53	59.7 ± 1.3	ESKD	Cut-off: BALP ≤ 27 U/l, osteocalcin 1≤; PTH: ≤ 150 pg/mL had the best specificity and sensitivity for detection of adynamic bone disease.
Salusky [17] 1988	USA	44 Female 22	11.8 ± 5.8	ESKD on CAPD	Bone formation rate correlated with serum PTH.
Sherard [38] 1993	USA and Canada	259 Female 63	PD: 60 ± 12 HD: 52 ± 1.4	ESKD	iPTH correlated directly with bone formation: higher in high bone turnover lesions, intermediate in mild lesions with normal bone formation
Torres [39] 1995	Spain (Canary Islands)	119 Female 80	47 ± 16	Predialysis 38 ESKD HD: 49 pts ESKD CAPD: 32 pts	iPTH level 120–250 pg/mL needed to avoid low bone turnover and HPT bone disease.
Monier-Faugere [15] 1996	USA	2248 Female 1144	Female 54 ± 0.4 Male: 50 ± 0.5	ESKD HD 81.4% CAPD18.6%	Serum ALP significantly greater in patients with HPT, and in low BTO. iPTH greater in pts with HPT than in those with MUO, and low BTO,, respectively.
Fletcher [123] 1997	USA	73		Dialysis	PTH > 100 pg/mL sensitivity of 81%, specificity of 66% for high BTO
Coen 2002 [124]	Italy	186		Predialysis: 79 pts HD: 107 pts	In dialysis pts, iPTH level of 150 pg/mL had a negative predictive value for low BTO of 96.4%, and a Youden index of 0.69. In predialysis pts. The Youden index was 0.57. AP had a Youden Index of 0.64.
Bervoets [58] 2003		84			For ABD, osteocalcin 41mcg/L, sensitivity of 83% and specificity of 67%. PPV 47%. Combination with BALP 23 U/L or less increased sensitivity, specificity, and PPV to 72%, 89% and 77%, respectively
Barreto [125] 2008	Brazil	97		ESRD on dialysis	For low turnover: PTH < 150 pg/mL, sensitivity of 50%, specificity of 85%, PPV 83% For high bone turnover: PTH >300 pg/mL, sensitivity 0.69, specificity 0.75, PPV 0.62
Lehman [126] 2005	USA	132		CKD 3–5	Patients with CKD stage 3–4 and low BTO had BI-PTH (biointact PTH) and iPTH (intact PTH) levels (in pg/mL ± SD) of 35 ± 34 and 59 ± 63. For high BTO, BI-PTH and iPTH 141 ± 60 and 221 ± 106, respectively. In CKD stage 5 and low BTO BI-PTH 51 ± 38 and iPTH 90 ± 60 pg/mL. For high BTO BI-PTH and iPTH levels of 237 ± 214 and 461 ± 437 pg/mL, respectively. Areas under the ROC curves for distinguishing low BTO from high BTO were 0.94 for BI-PTH and 0.91 for iPTH, respectively in stages 3–4. For CKD stage 5, values were 0.86 and 0.85, respectively.
Malluche [34] 2011	USA Europe	630 Female 301	55 ± 1	ESKD HD: 600 PD: 30	PTH not a significant predictor of bone turnover in black dialysis patients. PTH was a predictor of low bone turnover in white patients on dialysis
Haarhaus [127] 2015					PTH, BALP, Bix have similar diagnostic accuracy in distinguishing low from non-low BTO. BALP (AUC, 0.89) and PTH (AUC, 0.85) are useful for the diagnosis of non-low BTO. B1x can be used for the diagnosis of low BTO (area under the curve (0.83)
Sprague [35] 2016	Brazil 156 Portugal 89 Turkey 133 Venezuela 114	492 Female 218	49.5±	ESKD	iPTH and BALP allowed discrimination of low from non-low and high from non-high BTO. Optimal iPTH to discriminate low from non-low BTO: 104/pg/mL, and for high vs. non-high bone turnover: 323 pg/mL. Optimal BALP: 33.1 U/L, better for diagnosing low BTO. Combination of iPTH and BALP did not improve discrimination between low and high BTO
Marques [128] 2017	Brazil	31 Female 19	41 ± 11	ESKD HD 27, PD 3 No dialysis 1	Patients with low BTO: low PTH, BAP, and TRAP5b. BAP was the best predictor of BTO status. Best cut-off 66.6 U/L. Sensitivity 85, specificity 82.
Graciolli [47] 2017	Brazil	148 Female 51	50–54	CKD stage 2 -5	ROC PTH and FGF-23 curves were able to predict high BTO and low BTO. None of the markers was able to predict bone mineralization
Salam [44] 2018	UK	43	59 ± 12	CKD 4–5 including HD	BALP, intact PINP, and TRAP 5b better than iPTH for discriminating low from non-low BTO. iPTH can discriminate high BTO from non-high BTO
Lima [60] 2019	USA	104 Female 75	59 ± 15	CKD stage 2: 22 CKD stage 3: 29 CKD stage 4–519 ESKD on HD: 34	Serum activin increased with declining eGFR Activin showed similar AUC results, specificity, and sensitivity in predicting high turnover as iPTH, BSAP, and FGF-23 for discrimination of high vs. non-high bone turnover.

AUC: area under the curve; BALP bone alkaline phosphatase; B1x: isoform of BALP found in sermon of some patients with CKD; BBTO: bone turnover iPTH: intact parathyroid hormone; BI-PTH: biointact PTH; PPV: predictive positive value. ABD: adynamic bone disease. TRAP5b: tartrate-resistant acid phosphatase 5b.

## Data Availability

Not applicable.

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
