# Peer review of "Bone Disease in Chronic Kidney Disease and Kidney Transplant"

_nutrients, 2022, doi:10.3390/nu15010167_

Round 1

Reviewer 1 Report

The manuscript I was asked to review deals with the topic of bone disease in patients with CKD and after KTx.

It is well written and presents extensive insight into the discussed topic. Meteorically I have no major concerns - the manuscript is interesting and reads well. The only thing is it is very long and is hard to get through at in one reading. I would consider removing some parts that unnecessary double the presented topics - e.g. - lines 357 - 359 say "several studies have shown..." and then one of these studies is presented with much attention to detail in lines 360-364 - which is unnecessary. If a reader wants - she/he can easily find the study in references and just read it. This would make the manuscript easier to get through.

Still, this is just my point of view.

Kind regards

PS line 42 - I am clearly not a native English speaker, but I feel it should be "established" and not "stablished"

Author Response

Please, see file

Reviewer 2 Report

Dear Authors,

This is a very good and comprehensive review. No notes.

Author Response

Please, see attached file

Reviewer 3 Report

Authors present a review regarding bone disease in patients with renal problems both CKD and Kidney Transplant Recipients.

This is a good and interesting review with a particular focus on bone disease and its pathogenetic mechanisms. Some major revisions are needed before the publication of this article.   

In the section “Bone disease in CKD” authors described different bone pattern in CKD patients.

-    At page 2 line 67-67, authors reports “More recently, it has been shown that osteoporosis is a frequent feature in patients with CKD-MBD”. After this sentence, authors should define the concept of osteoporosis in general population and CKD population. Please specify.

-       Authors reports in an exhaustive way different studies among the proportion of bone turn-over status according to the renal function and CKD stage. They could improve this section with a new summary table.

-       At page 6, authors reports that there are “other factors such as gender, coexisting disease, age, ethnicity … responsible for the diversity bone lesions”. They should cite drugs too (e.g. corticosteroids).

-       At page 6 line 196-199 authors reports: “In a recent reanalysis of two sets of data of bone histomorphometry in kidney transplant patients using reference ranges from different laboratories, there were discrepancies that at the end may alter the classification of the type of renal osteodystrophy in particular patients”. This sentence is not clear: why does authors cite KTRs in this section? Please specify.

In the section “Pathogenesis of bone disease in CKD-MBD” authors described the different pathogenetic mechanisms. They should include a summary figure in this section in order to reduce the too extensive description.

In the section “Bone Fracture in CKD” authors described in a very clear way the problem of bone evaluation in CKD patients: they do not cite the possibility to evaluate bone microarchitecture thanks to trabecular bone score. Do the authors have any information about it?

In the section “Bone Disease after Kidney Transplantation” authors should explain in a better way the difference between persistent and de novo hyperparathyroidism after kidney transplantation and the (Cianciolo G, FGF23 in kidney transplant: the strange case of Doctor Jekyll and Mister Hyde CKJ 2016, Cianciolo G A road map to parathyroidectomy for kidney transplant candidates, CKJ 2022). Please specify.

At the same time, they do not speak about the possible role in pathogenetic mechanisms of different drugs (not only corticosteroids but CNIs too). Please specify.

Author Response

Please, see attached file

Reviewer 4 Report

The authors performed an extensive review on Bone Mineral Disease in CKD and after kidney transplantation. The review is very detailed but some substantial changes are needed to make it more usable.

On page 3, Table 1 does not indicate the reference GFR of CKD in the indicated studies. It is unclear whether the level of BMD is comparable for GFR values. Also, on page 3 line 94 we talk about advanced CKD without specifying the GFR value and the ongoing therapies. On page 3 line 111 Malluche's work is cited as incipient kidney disease. The level of GFR is unclear. Furthermore, from a logical point of view, it is easier to quote first the authors who refer to early stages of CKD and then the authors who refer to the advanced stages, mentioning however which type of CKD stage. On page 4 lines 114 to 133 the text is confused. It is said first that low turnover disease is the most frequent bone disease in the early and late stages of CKD (line 117), then it is said that low turnover is the most frequent form of bone disease in the early stages of CKD (line 130). Finally, it is said that low turnover is the most frequent form in patients with moderate form of CKD. It is necessary to modify the text more clearly and possibly collect the authors in a new table, always specifying the stages of CKD to which reference is made.

On page 4 from line 135 to line 160 reference is made to an abstract of the authors which is described too fully and to which 2 of the 3 figures of the article refer. It should be mentioned in the summary table. On page 5 line 192 it is said that various factors contribute to the histological pattern of bone mineral disease during CKD, strangely, therapy with for example vitamin D is not mentioned (Cianciolo G, et al. Effect of vitamin D receptor activator therapy on vitamin D receptor and osteocalcin expression in circulating endothelial progenitor cells of hemodialysis patients. Blood Purif. 2013). The action of heparin administered during dialysis on bone metabolism is not mentioned (Cianciolo G, et al. Effects of unfractionated heparin and low-molecular-weight heparin on osteoprotegerin and RANKL plasma levels in hemodialysis patients. Nephrol Dial Transplant. 2011). In the section "Pathogenesis of bone disease in CKD-MBD" the lines between 206 and 217 must be summarized in an explanatory figure. Pages 7 and 8: a new table should indicate the functions of FGF23, PTH and sclerostin to streamline the text, indicating the roles demonstrated and presumed for each factor.

It seems to me that the role of the Wnt / beta-catenin pathway can be omitted given the lack of robustness of the data in favor, in the same way the role of indoxyl phosphate.

In the section "Bone fracture CKD" it is useful a table indicating the epidemiology of the problem in the various stages of CKD. Lines 374-382: osteoporosis. The description of the characteristics of osteoporosis does not go here but in the section "bone disease in CKD". Page 10. It would be advisable to have a table indicating what the DEXA is for and what the HR-pQCT is for. Including costs for each exam. Is it possible to insert an image of a pathological DEXA and a normal one? Is it possible to insert an image of normal HR-pQCT and a pathological HR-pQCT?

Page 11. We need a table with the biomarkers with certain role and those with presumed role. Line 486 "alterations in mineral metabolism seem of primary importance in the development of post-transplant bone disease" The sentence is obvious and must be deleted, rather the causes that can lead to metabolic alterations including the extent of immunosuppressive therapy and in particular the steroid therapy burden.

Figure 4 needs to be corrected: CKD-MBD does not appear in the center of the red circle. Then, particularly the arrow between renal osteodystrophy and osteoporosis probably needs to be corrected in both directions

Author Response

Please, see attached file

Round 2

Reviewer 3 Report

The changes that have been made are acceptable. Therefore the article is liable to publication in that state.

Author Response

Dear Reviewer:
Thank you very much for your comments and suggestions to our manuscript. We appreciate very much
all the effort you have put in a thorough review our work. Your comments have been addressed as follows
and the corresponding modifications have been done in the new version of the manuscript.
1. Reviewer comment: Table 1 and 2 lack reference to CKD-MBD therapy for most authors. One
cannot speak of bone turnover without understanding whether this turnover is of iatrogenic origin
or not. In particular, we know that high dose of vitamin D or calcimimetics may influence bone
turnover
Response: Thank you for your comment. We agree that these are important aspects to consider
while evaluating the bone turnover in the different studies reported. Tables 1 and 2 have been
updated and extended. Now they content information about use of vitamin D analogs, phosphate
binders, and calcimimetics as well as past history of parathyroidectomy or kidney transplant when
available. These information are also commented in the text, lines 123 to 128 and 161 to 181.
2. Reviewer comment: It is not clear why a high turnover disease is more likely in historical patients
while a low turnover disease is more recently describe
Response: These aspects are also addressed in connection with the comments mentioned above.
3. Reviewer comment: In both table 1 and 2 there are several empty boxes.
Response: We apologize for the empty boxes in tables 1 and 2. This has been corrected in the new
versions of the tables included in the revised document.
4. Reviewer comment: Line 144 (page 4): The sentence needs to be rewritten; it is not clear. The
authors do not clarify whether in their experience reported by the Abstract the turnover is
influenced by the current therapy for mineral metabolism.
Response: Patients with CKD stages 3 to 5 in our study were not receiving therapy that may
influence mineral metabolism. This information was added to the text.
Figure 2 was deleted as the number of patients for the different categories of CKD stages 3 to 5
was relatively low as to draw strong conclusions. In the new version of the manuscript, numbers
were pooled for CKD stages 3 to 5 as a group. The information is now included in the text, lines
149 to 161. Additional discussion was added in lines 169 to 172.
5. Reviewer comment: In the legend to figure 2, it would be important to include the number of
patients for each of the CKD stages.
Response: Please see response to comment number 4 above.
6. Reviewer comment: The new figure 3 and 4 were not attached to the new version of the
manuscript and therefore I could not see them.
Response: All the four figures of the paper were sent in a different file. We are very sorry that the
reviewer did not received them. As mentioned above, figure 2 of the previous version of the
manuscript was eliminated. In the current revised version, there are 3 figures in total. All 3 figuresare sent as a separate file to avoid trouble with formatting the manuscript. We expect the editorial
office do the corresponding inclusion of them in the manuscript.
7. Reviewer comment: Page 7 and 8: A new table should indicate the functions of FGF-23, PTH and
sclerostin to streamline the text, indicating the roles demonstrated or presumed for each factor.
Response: Thank you for your suggestion to include a new table indicating the roles demonstrated
and presumed for FGF-23, PTH and sclerostin. However, we consider that this goes beyond the
scope of the present paper.
8. Reviewer comment: In the section “Bone fracture in CKD” it is useful a table indicating the
epidemiology of the problem in the various stages of CKD.
Response: A new table (Table 3. Incidence of fractures in CKD) was added to the new revised
version of the manuscript. This table summarizes several published studies examining the
incidence of fractures in different populations of CKD patients. It is important to clarify that many
of the studies follow a different methodology of investigation and analysis of the results, thus
limiting the comparison between studies. Therefore, a comment column was added to the table
to expand the information about the results. Please, see lines 316 to 323, and 327 to 346.
Point # 9 was responded with # 8. We defer the decision to include this table as supplementary
material to the reviewer and the editorial office.
Reviewer comment: We need a table with biomarkers with certain role and with presumed role
Response: A new table (Table 4. Relationship between bone markers and discrimination of the
type of bone turnover alteration in CKD.) with reported relationship between biomarkers and
diagnosis of bone turnover was added. Discussion of the findings reported in the table is provided
in lines 425 to 426, and lines 437 to 449.
We have performed a detailed revision of the manuscript and addressed most of the comments of the
reviewer. We are thankful to reviewer as we feel that his comments and suggestions have helped
improving the manuscript.
We hope the manuscript is now suitable for publication.

Reviewer 4 Report

The authors did not fully respond to the reviewer's comments, and still need to clarify some points.

1)      Both Table 1 and Table 2 lack references to CKD-MBD therapy for most authors. One cannot speak of bone turnover without understanding whether this turnover is of iatrogenic origin or not. In particular we know that high doses of vitamin D or more recently therapy with calcium mimetics can influence bone turnover, as well as protracted therapy with aluminum-based phosphate binders. If such information is not available, it must be defined in the tables.

2)      It is not clear why a high-turnover disease is more likely in historical patients while a low-turnover disease is more recently described. The authors must clarify this passage on the basis of considerations already present in the literature or on their own. Is it the age of the patients to blame? Or the excess of CKD-MBD therapy? The absence of a role for vitamin D was reported by Malluche, but what did the other authors find?

3)      In both Table 1 and Table 2 there are several empty boxes. They concern the age of the patients related to the works cited, the renal function, the percentage of patients with high or normal turnover, the nationality of the authors. This is important information. If the authors have not been able to obtain them from reading the articles, it must be specified in the legend.

4)      Line 144 (page 4): the sentence needs to be rewritten; it is not clear. The authors do not clarify whether in their experience reported by the Abstract the turnover is influenced by the current therapy for mineral metabolism.

5)      In the legend to figure 2 it would be important to include the number of patients for each of the CKD stages.

6)      The new figure 3 and the figure 4 were not attached to the new version of the manuscript and therefore I could not see them.

7)      Pages 7 and 8: a new table should indicate the functions of GGF-23, PTH and sclerostin to streamline the text, indicating roles demonstrated and presumed for each factor. The new table would also be useful for shortening the manuscript. Alternatively, it can be included in the supplementary material.

8)      In the section “Bone fracture in CKD” it is useful a table indicating the epidemiology of the problem in the various stages of CKD”.

9)      The table is useful for a reader who has to deal with 22 pages on such a difficult topic. Alternatively, the new table can be included in the Supplementary Material.

10)   Page 11. We need a table with biomarkers with certain role and with presumed role. I confirm what I wrote above.

Author Response

Manuscript: Nutrients 1994086 Revised version 12/16/2022
Response to Reviewer 4.
Dear Reviewer:
Thank you very much for your comments and suggestions to our manuscript. We appreciate very much
all the effort you have put in a thorough review our work. Your comments have been addressed as follows
and the corresponding modifications have been done in the new version of the manuscript.
1. Reviewer comment: Table 1 and 2 lack reference to CKD-MBD therapy for most authors. One
cannot speak of bone turnover without understanding whether this turnover is of iatrogenic origin
or not. In particular, we know that high dose of vitamin D or calcimimetics may influence bone
turnover
Response: Thank you for your comment. We agree that these are important aspects to consider
while evaluating the bone turnover in the different studies reported. Tables 1 and 2 have been
updated and extended. Now they content information about use of vitamin D analogs, phosphate
binders, and calcimimetics as well as past history of parathyroidectomy or kidney transplant when
available. These information are also commented in the text, lines 123 to 128 and 161 to 181.
2. Reviewer comment: It is not clear why a high turnover disease is more likely in historical patients
while a low turnover disease is more recently describe
Response: These aspects are also addressed in connection with the comments mentioned above.
3. Reviewer comment: In both table 1 and 2 there are several empty boxes.
Response: We apologize for the empty boxes in tables 1 and 2. This has been corrected in the new
versions of the tables included in the revised document.
4. Reviewer comment: Line 144 (page 4): The sentence needs to be rewritten; it is not clear. The
authors do not clarify whether in their experience reported by the Abstract the turnover is
influenced by the current therapy for mineral metabolism.
Response: Patients with CKD stages 3 to 5 in our study were not receiving therapy that may
influence mineral metabolism. This information was added to the text.
Figure 2 was deleted as the number of patients for the different categories of CKD stages 3 to 5
was relatively low as to draw strong conclusions. In the new version of the manuscript, numbers
were pooled for CKD stages 3 to 5 as a group. The information is now included in the text, lines
149 to 161. Additional discussion was added in lines 169 to 172.
5. Reviewer comment: In the legend to figure 2, it would be important to include the number of
patients for each of the CKD stages.
Response: Please see response to comment number 4 above.
6. Reviewer comment: The new figure 3 and 4 were not attached to the new version of the
manuscript and therefore I could not see them.
Response: All the four figures of the paper were sent in a different file. We are very sorry that the
reviewer did not received them. As mentioned above, figure 2 of the previous version of the
manuscript was eliminated. In the current revised version, there are 3 figures in total. All 3 figures
